# Pancreatic Cancer and Detection Methods

**DOI:** 10.3390/biomedicines11092557

**Published:** 2023-09-18

**Authors:** Umbhorn Ungkulpasvich, Hideyuki Hatakeyama, Takaaki Hirotsu, Eric di Luccio

**Affiliations:** Hirotsu Bioscience Inc., 22F The New Otani Garden Court, 4-1 Kioi-cho, Chiyoda-ku, Tokyo 102-0094, Japan; umbhorn@hbio.jp (U.U.); h.hatakeyama@hbio.jp (H.H.); hirotsu@hbio.jp (T.H.)

**Keywords:** *C. elegans*, N-NOSE, early-cancer screening, pancreatic cancer, detection methods

## Abstract

The pancreas is a vital organ with exocrine and endocrine functions. Pancreatitis is an inflammation of the pancreas caused by alcohol consumption and gallstones. This condition can heighten the risk of pancreatic cancer (PC), a challenging disease with a high mortality rate. Genetic and epigenetic factors contribute significantly to PC development, along with other risk factors. Early detection is crucial for improving PC outcomes. Diagnostic methods, including imagining modalities and tissue biopsy, aid in the detection and analysis of PC. In contrast, liquid biopsy (LB) shows promise in early tumor detection by assessing biomarkers in bodily fluids. Understanding the function of the pancreas, associated diseases, risk factors, and available diagnostic methods is essential for effective management and early PC detection. The current clinical examination of PC is challenging due to its asymptomatic early stages and limitations of highly precise diagnostics. Screening is recommended for high-risk populations and individuals with potential benign tumors. Among various PC screening methods, the N-NOSE plus pancreas test stands out with its high AUC of 0.865. Compared to other commercial products, the N-NOSE plus pancreas test offers a cost-effective solution for early detection. However, additional diagnostic tests are required for confirmation. Further research, validation, and the development of non-invasive screening methods and standardized scoring systems are crucial to enhance PC detection and improve patient outcomes. This review outlines the context of pancreatic cancer and the challenges for early detection.

## 1. The Pancreas

### 1.1. Structure of the Pancreas

The pancreas is a visceral organ located posterior to a stomach in the form of a leaf. It measures roughly 15 cm in length and 5 cm in width in adults. The pancreas is subdivided into four distinct regions: a head, neck, body, and tail. The head of the pancreas is the broader part, situated adjacent to the stomach and duodenum junction, where the bile duct either lies within a groove on its surface or passes through its substance. The neck and body regions are surrounded by major blood vessels. Finally, the tail of the pancreas is located anterior to the left kidney and spleen [1], as shown in Figure 1a.

### 1.2. Function of the Pancreas

The organ comprises both endocrine and exocrine cells, which perform distinct functions. The majority of pancreatic cells are exocrine cells, which make up the exocrine glands and ducts. These cells are responsible for secreting a range of enzymes including amylase for carbohydrate digestion, proteases such as trypsin, chymotrypsin, elastase, and carboxypeptidases for amino acids digestion, and lipase for the breakdown of fatty acids. In contrast, the endocrine cells comprise a relatively small proportion of the organ but play a crucial role in the secretion of hormones such as insulin and glucagon, which regulate blood glucose levels by releasing them directly into the bloodstream. The pancreas primarily comprises exocrine cells forming the exocrine glands and ducts [1,2].

### 1.3. Disease of the Pancreas

The pancreas is an enigmatic organ whose serious pathologies manifest acutely or chronically. This is partly due to its anatomical location, which renders it obscured by other visceral structures and thus challenging to image, even with digital imaging modalities.

#### Pancreatitis

Among the most prevalent pancreas diseases is acute pancreatitis, an inflammatory condition triggered by factors such as alcohol consumption, gallstones, type 2 diabetes, smoking onset, and hypertriglyceridemia [3,4] (see Figure 1b). The incidence of acute pancreatitis varies widely, with estimates ranging from 10–40 cases per 100,000 population, with a tendency towards higher rates in regions with high alcohol consumption [4]. Gallstones, which may dislodge the gallbladder and obstruct the bile duct, are another common cause of acute pancreatitis, particularly in the elderly population. The obstruction of the bile duct and pancreatic duct, which converge at the ampulla of Vater, leads to pancreas inflammation. Additionally, type 2 diabetes has been associated with an elevated risk of gallstone [3].

Smoking is associated with tissue inflammation and fibrosis, when coupled with heavy alcohol consumption, can incite severe inflammation, rendering pancreatic cells more susceptible to autodigestion [5,6]. Hypertriglyceridemia is characterized by an elevated serum fatty acid concentration, which can inhibit mitochondrial injury in pancreatic acinar cells, culminating in inflammation and a consequential reduction in pancreatic duct function [7].

Acute pancreatitis can be a serious condition, depending on the patient’s age. The mortality rate is relatively low (~1%) in young patients and significantly higher (~30%) in older patients [8].

Minor developed pancreatitis can be the result of hereditary or familial predisposition. This disease subtype is caused by genetic mutations, specifically in the trypsinogen gene (PRSS1) encoding for cationic trypsinogen (a protein digestion-related enzyme). The transmission of this genetic variant follows an autosomal dominant pattern, with an estimated penetrance of 80%, meaning individuals who carry the mutation have an 80% chance of developing pancreatitis during their lifetime [9,10]. Moreover, the mutant carrier has an increased risk of developing chronic pancreatitis, estimated to be 30–40%, where the incidence of chronic pancreatitis in the general population is around 1–2% [8]. Moreover, a small percentage of acute pancreatitis patients have a chance of developing PC (2%) [11].

On the other hand, chronic pancreatitis is characterized by the gradual and irreversible degeneration of the pancreas, caused by persistent inflammation, which destroys both exocrine and endocrine cells [12]. It is associated with a significantly elevated risk of developing pancreatic cancer, roughly 20 times higher than the general population [13]. In certain cases, there are limited therapeutic options, such as pancreatectomy.

To diagnose pancreatitis, a healthcare provider will examine the patient for symptoms such as sudden and severe abdominal pain, nausea and vomiting, and fever, and also consider the patient’s medical history, particularly any history of alcohol consumption. To confirm a diagnosis of acute pancreatitis, a healthcare provider typically looks for two out of three criteria: (1) abdominal pain consistent with pancreatitis, (2) a blood test showing elevated levels of enzymes serum amylase or lipase that is three times higher than normal, and (3) imaging such as computed tomography (CT) or magnetic resonance image (MRI), or transabdominal ultrasound (TUS) that shows consistent signs of pancreatitis [14,15]. This highlights that both pancreatitis and PC can share similar symptoms, such as abdominal pain and weight loss [16], but PC often has additional symptoms, such as jaundice [17], which are not typically present in pancreatitis and symptoms tend to be more persistent.

## 2. Pancreatic Cancer (PC)

### 2.1. Incidence of Pancreatic Cancer

PC represents a frequent contributor to oncological mortality and is distinguished by a high degree of variability, a heavily populated stromal microenvironment within the tumoral mass, an inclination towards extensive metastasis, and a significant reconfiguration of cellular metabolism (see in Figure 1b).

Within the realm of PC, malignancies can arise from either exocrine or endocrine cells. The most prevalent form of PC originates from exocrine cells, specifically known as pancreatic ductal adenocarcinoma (PDAC). PDAC accounts for 90% of the total PC and it is a malignancy that has a dismal outcome, with a 5-year overall survival rate of less than 10%, even though the development of the medical treatment [18]. This PC is caused by the abnormal growth of exocrine cells, accounting for over 90% of all pancreatic malignancies [19].

Among the exocrine cancers, a subtype is acinar cell carcinomas, characterized by their abnormal secretion of enzymes. Furthermore, there are fewer common types of exocrine PC, such as adenosquamous carcinomas, squamous cell carcinomas, signet ring cell carcinomas, undifferentiated carcinomas, and undifferentiated carcinomas with giant cells. However, these subtypes are much less common and account for a smaller percentage of PC cases [20]. Conversely, the remaining PC subtypes develop improperly from endocrine or neuroendocrine cells, a phenomenon intricately linked to hormone production [21].

Despite being a relatively uncommon cancer among all malignancies, PC is a leading cause of cancer-related mortality. According to the Globocan 2020 report, it ranks as the 12th most frequent cancer incidence (4.96 million cases), among the top 7 leading causes of cancer-related deaths (4.66 million deaths). The report indicates that there is a higher prevalence of PC among men in comparison to women. Furthermore, this malignancy is more prevalent in countries with high human index (HDI) compared to low HDI countries, and there has been a slight upward trend in incidence and death for two decades [22,23,24,25,26].

### 2.2. Risk Factors

Investigating the risk factors for PC can be a valuable approach to preventing the disease, despite the current lack of clear etiology. The susceptibility to the disease is possibly influenced by inherited and modified factors. In Figure 2, several notable associations have been concerned regarding inherited factors, such as age, gender, ABO blood group, ethnicity, microorganism, family history, and underlying conditions such as diabetes. These factors may play a role in the development and progression of PC.

#### 2.2.1. Ethnicity

In the United States (US), most PC new cases and deaths occur in individuals aged 55 years and older, with a combined peak incidence rate of 89.6% and a combined peak mortality rate of 93.3% in 2022 [27]. The median age at diagnosis is 70 years for new cases and 72 years for deaths. Furthermore, it is worth noting that the incidence and mortality rates for individuals under 40 are relatively low, at 96.5%, compared to other countries such as China, which highlights the trend that older individuals are more susceptible to PC [28]. Although PC primarily affects older groups, early detection and screening measures may decrease the incidence and mortality rates by detecting the disease in younger individuals. Demographic analysis of new cases in the US by ethnicity and gender reveals that the male population has a higher rate of occurrence than the female population (0.15 million cases in men and 0.11 million cases in women) across all ethnic groups, with the highest incidence in the black, Indian Alaska Native, and white groups, followed by the Hispanic group, and the lowest incidence in the Asian/Pacific Islander group [27].

#### 2.2.2. Genetics and Epigenetics

The genomic perturbation in the inception and advancement of PC encompasses genetic expression modifications, chromosomal anomalies, and epigenetic transformations, leading to its manifestation. The amassment of genetic mutations in the precancerous lesions in the pancreas, referred to as Pancreatic Intraepithelial Neoplasia (PanIN), culminates in its degradation into PC. An additional risk factor for PC is a hereditary or familial component accounting for approximately 10% of all PC cases [29,30]. Healthy family members should seek genetic counseling if there is a first-degree relative with early-onset PC (<50 years) or multiple first-degree relatives with PC [31].

Most PC sufferers exhibit mutations in the proto-oncogene Kirsten rat sarcoma viral oncogene homolog (KRAS) in approximately 90% of all PC cases [32,33]. These mutations impede GTPase functionality and disrupt other signal transduction pathways, resulting in abnormal cellular behavior. Despite being recognized as a marker for PC, KRAS alone has been deemed inadequate for fully explaining PC development [34]. Inactivation of the Tumor suppressor protein p53 (TP53) due to mutation is present in 50–74% of PC cases and leads to impaired DNA recognition and obstruction of cell cycle arrest [35,36]. The occurrence of additional mutations in key tumor suppressor genes, including cyclin-dependent kinase inhibitor 2A (CDKN2A), SMAD family member 4 (SMAD4), has been linked to the disruption of the cell cycle and the eventual development of carcinogenesis in 46–60% of CDKN2A cases [35,36,37] and 31–38% of SMAD4 cases during the advanced stages of PC [36,38].

Epigenetic changes refer to modifications to DNA that impact gene expression without changing the underlying DNA sequence. These changes are often associated with specific genes related to histone modification enzymes. The histone modification enzymes, including mixed-lineage leukemia (MLL2/3) histone methylases, lysine demethylase 6A (KDM6A), and histone methyltransferase, are related to histone modification and chromatin-regulating genes. These alterations promote uncontrolled cell growth and contribute to PC development [34,39,40].

#### 2.2.3. Blood Group

In addition, an established risk phenotype is individuals with blood group A. Several studies have revealed that individuals with blood group A have a significantly increased risk of developing PC compared to all other ABO blood types. The ABO blood group system is determined by the presence of antigens on the surface of red blood cells and is controlled by the ABO gene. The A antigen is produced by the A1 allele and A2 allele. The A1 allele is known to make more of the A antigen than the A2 allele and thus is thought to increase the risk of PC more than the A2 allele. It has been found that the presence of B antigen is associated with an increased risk of PC compared to individuals with blood type O. Conversely, individuals with blood group O have been found to have a decreased risk of PC, as they do not possess the antigen and therefore do not produce the enzyme [41]. It is also worth noting that ABO blood type may influence PC risk by modulating dietary inflammation, such as by increasing the risk of PC through high-fat and high-calorie diets [42].

#### 2.2.4. Diabetes and Obesity

Diabetes patients have a higher risk of developing PC, and in 80% of diabetes patients, developing PC can be detected in the pre-symptomatic phase [43]. This is possibly due to chronic hyperinsulinemia (high levels of insulin in the blood), which can lead to the growth and proliferation of cancer cells. Therefore, individuals with a certain blood type A and diabetes may have an even greater risk of developing PC [44,45]. Moreover, obesity and a high-calorie diet are related to a pro-inflammatory condition in PC due to inflammation and hormonal imbalances and are often associated with insulin resistance and diabetes [46]. Some researchers pointed out that obesity can cause the activation of KRAS, a protein that plays a significant role in PC development. Thus, obesity and a high-fat diet are proposed inducers of tissue inflammation in the development of the PC [47,48].

#### 2.2.5. Smoking and Alcohol

As an avoidable risk factor, studies suggest that smoking has a significant role in the development of PC, with a 72% increased risk for current smokers compared to non-smokers [49] and that smoking-related carcinogens such as nicotine, benzo(a)pyrenes, and tobacco-specific nitrosamines play a role in the development of the disease by forming DNA adducts and leading to mutations in key genes [50,51]. The overall burden of PC could be reduced by 25% if smoking was eliminated [52]. Moreover, heavy alcohol consumption is associated with a higher risk of PC. Increased accumulation of acetaldehyde generated in alcohol metabolism in humans may accelerate the progression of tumors by boosting pancreatic inflammation [53,54]. A genetic polymorphism in the ALDH2 enzyme can also increase the risk of developing alcohol-related cancers in individuals carrying the ALDH2 × 2 allele in the East Asian population [55]. The meta-analysis indicated a 20% elevated risk of PC among individuals who consumed three drinks per day (equivalent to 37.5 g of ethanol) when compared with non- or infrequent drinkers. In contrast, the relative risk did not increase significantly among light or moderate alcohol drinkers [56].

#### 2.2.6. Microbiota

Moreover, several studies have demonstrated an unambiguous correlation between human microbiota and PC development [57,58], particularly regarding gastrointestinal and pancreatic microbes, as well as viral infections such as hepatitis and bile [59,60]. The existence of the microbes within the pancreas exacerbates PC development through its promotion of innate and adaptive immune suppression [61] and the production of secondary bile acids, lipoteichoic acid, and short-chain fatty acids [62]. Additionally, microbiota dysbiosis has also been shown to be a contributing factor in PC. Those with PC have been identified as exhibiting a deficiency of *Neisseria elongata* and *Streptococcus mitis* exists in the oral cavity [63], an elevated ratio of *Leptotrichia* genus to *Prophyromonas* genus in the saliva [64], and the presence of *Enterococcus faecalis* in pancreatic tissue [65].

### 2.3. Metastasis of PC

As numerous studies have acknowledged, PC is regarded as a highly virulent tumor, demonstrating a remarkable predisposition towards metastasis. The neoplastic cells, which secede from the primary lesion, traverse through the blood circulatory and lymphatic system, ultimately reaching the liver, which is the most frequent destination for metastasis, with the lungs, peritoneum, and bones following closely [66,67,68]. The tumor carriers, known as exosomes, originating from PC cells, possess the capability to trigger the cancer cells to undergo epithelial-to-mesenchymal transition (EMT) and secrete exosomes that enhance the tumor microenvironment culminating in liver metastasis [69,70]. Consequently, early screening is the most efficacious method of preventing this pernicious cancer. The necessity of screening for PDAC patients is suggested by various risk factors such as family history, chronic pancreatitis, and behavior, which may play a role in the onset of the condition. However, solely relying on these risk factors is inadequate for definitively confirming the presence of cancer. To ensure accuracy, a combination of clinical indicators, including abdominal and back pain, weight loss, jaundice, type II diabetes mellitus, alterations in enzyme levels, and results from LB tests, along with imaging analysis, must be considered.

## 3. Detection Investigations

### 3.1. Current Examination Circumstance

The asymptomatic nature of PC in its initial stages often leads to a delayed diagnosis until the disease has progressed to a more advanced stage, resulting in a significantly reduced survival rate and affecting the efficacy of current therapeutic options. On the other hand, the mortality rate of other cancers, such as breast cancer, has seen a substantial decline of 43% between 1989 and 2020 due to advancements in diagnostic and screening procedures. Consequently, the importance of early detection in enhancing the prognosis of PC cannot be overstated [71]. Conducting a targeted examination of individuals for PC within the general population is arduous and financially unfeasible due to the inadequacy of highly precise diagnostic evaluations and the low incidence of PC. Therefore, the current screening for PC is limited to populations with potential benign tumors and those who fit the high-risk criteria recommended by the International Pancreatic Cancer Screening Consortium and the U.S. Prevention Services Task Force (USPSTF) [72,73,74]. Concerning the clinical presentation of PC, it is common for the afflicted individual to experience conspicuous symptoms such as weight loss, pain, cutaneous indications (such as jaundice), and appreciable mass, which may not reveal any abnormalities during a physical examination [75].

In 2020, the American Gastroenterological Association (AGA) issued comprehensive guidelines for PC screening, with a specific focus on individuals at high risk. These high-risk groups encompass individuals afflicted with conditions such as Peutz–Jeghers syndrome, CDKN2A gene mutation, hereditary pancreatitis, Lynch syndrome, or those with a first-degree (or more) family history of PC. Furthermore, individuals with mutations in BReast CAncer gene 1 and 2 (BRCA1, BRCA2), partner and localizer of BRCA2 (PALB2), and Ataxia–Telangiesctasia Mutated (ATM) genes are also classified as high-risk candidates. Age-specific recommendations advise individuals above 50 or below 10 years of age with familial onset to begin PC screening protocols. For those at the age of 40 with CDKN2A and PRSS1 gene mutation associated with high-risk conditions, a PC screening regimen is proposed. This entails annual PC screenings for individuals with non-PC results and a more frequent 3–6-month screening schedule for those with newly suspected PC. The AGA advocates for a multi-modal screening approach, which combines advanced imaging techniques such as a combination of magnetic resonance imaging (MRI) and endoscopic ultrasound (EUS) and the assessment of tumor markers [76]. Nevertheless, it is noteworthy that several screening/diagnostic methodologies (as shown in Figure 3) are available to both the general population and high-risk individuals, extending beyond the techniques mentioned earlier.

### 3.2. Imaging Modalities

Imaging techniques have been employed for PC screening patients with clinical symptoms and associated risk factors. A comprehensive imaging assessment, which includes non-invasive methods such as ultrasound, computed tomography (CT), MRI, positron emission tomography (PET), endoscopic retrograde cholangiopancreatography (ERCP), and EUS, is recommended for screening purposes. However, specific imaging modalities may be inadequate in detecting early lesions or distinguishing benign from malignant lesions. Therefore, selecting the appropriate imaging technique depends on the patient’s overall health, symptoms, and medical history [77].

#### 3.2.1. Ultrasound

Ultrasound is a non-invasive, cost-effective diagnostic tool frequently used in individuals with jaundice or abnormal pain. Although its accuracy is estimated to be between 50–70%, its sensitivity and accuracy in detecting PC are controversial, with factors such as operator experience, size of the tumor, patient body habitus (adipose tissue), and presence of bowel gas influencing the quality of the image [78].

Although traditional ultrasound has limitations, EUS produces high-quality images and has become a valuable diagnostic tool. EUS uses endoscopy and ultrasound imaging to generate detailed images of the pancreas, even in the case of small tumors (<3 cm) [79,80], without the risk of ionizing radiation associated with CT scans [73]. However, the conventional EUS alone may only sometimes be sufficient, despite its high sensitivity and specificity over cross-sectional imaging fields [81,82]. Therefore, EUS is frequently combined with other methods, such as contrast-enhanced (CE), elastography, fine-needle aspiration (FNA), and fine-needle biopsy (FNB), to improve PC evaluation and avoid misdiagnosing it as pancreatitis.

CE-EUS is a technique that employs high-resolution endoscopic ultrasound waves and intravenous contrast to aid in identifying pancreatic lesions [83]. It is a highly accurate method with a sensitivity of 91% and specificity of 93% [84,85], that can effectively differentiate between lesions caused by pancreatitis and those resulting from PC [54,86]. EUS-elastography is a non-invasive method that employs an ultrasound probe to evaluate the soft tissue’s elasticity. Used in conjunction with endoscopic ultrasound fine-needle aspiration (EUS-FNA), this technique can assist in distinguishing between malignant and benign lesions by measuring the velocity of a shearing wave as it passes through the tissue. With a sensitivity of 95–97% and specificity of 67–76%, this technique is suitable for diagnosing PC lesion fields [74,79,80,81], and is becoming another gold standard for PC diagnosis [87].

While the endoscopic procedure with the ultrasound probe is employed for tissue evaluation, ERCP integrates endoscopic techniques with X-ray imaging for diagnostic purposes in bile and pancreatic ducts distinct from the ultrasound-guided sample collection [82]. Additionally, alternative non-invasive imaging methods, including CT and MRI scans, offer superior image resolution for the assessment of pancreatic lesions. These imaging techniques provide enhanced detail and precision in evaluating the pancreatic tumors [82]. Consequently, CT and MRI scans are often considered for PC diagnosis as well.

#### 3.2.2. CT

A conventional CT scan generates sequential three-dimensional images using a rotational and continuous X-ray technique. This method assists in detecting lesions by enhancing the contrast (CT density) between normal tissue and abnormal tumors. Furthermore, multidetector CT (MDCT) presents higher-resolution images and quicker imaging duration compared to conventional CT scans [88,89].

MDCT scans are the preferred comprehensive imaging modality for patients at high risk for PC. This type of CT scan is performed in two phases, an arterial phase and a portal phase, producing hypodense images that improve the detection sensitivity (75–100%) and specificity (70–100%) of the examination [90,91]. The accuracy and sensitivity of CT scans in detecting PC tumors greater than 2 cm in size are high, with a detection accuracy of 73% and sensitivity of 69% for smaller lesions [91].

Regarding post-therapeutic assessment, PET/CT is frequently utilized alongside MDCT. PET/CT is a molecular imaging modality that utilizes the radiotracer fluorine 18-fluorodeoxyglucose (FDG) to identify PC with a reported level of sensitivity ranging from 46% to 71% and specificity from 63% to 100% [89,92]. PET/CT in monitoring treatment post-chemotherapy or radiation is a common practice and has demonstrated efficacy [89,93].

#### 3.2.3. MRI

MRI has been employed in identifying pancreatic neoplasms when results from ultrasound or CT are equivocal, emphasizing characterizing cystic lesions that cannot be discerned through CT scans. MRI’s imaging quality and diagnostic accuracy have been significantly improved with recent advancements in MRI scanners and techniques, resulting in enhanced soft-tissue contrast compared to CT scans [94]. As a result, MRI is now a valuable tool for patients, particularly in detecting small tumors and metastases. During the pancreatic and venous phases of an MRI examination, PC appears hypointense on T1-weighted images that are enhanced with a gadolinium agent, owing to the hypo-vascularity of cancer, accompanied by abundant fibrous stroma in comparison to the surrounding pancreatic parenchyma [89,95]. The sensitivity and specificity of MRI studies are in the range of 81–99% and 70–93%, respectively [83,96].

Magnetic resonance cholangiopancreatography (MRCP) is a type of MRI that is particularly useful in detecting narrowing in ductal systems and stones as alternate causes of biliary or pancreatic ductal dilatation. The sensitivity of MRCP is between 85–87%, while the specificity is between 93–95%. MRCP enables early and successful detection of tumors and analysis of morphological changes within the pancreas [88]. In addition to imaging, conducting a biopsy investigation can corroborate staging outcomes and enhance imaging-based diagnoses by integrating biological PC biomarkers.

### 3.3. Tissue Biopsy

Tissue biopsy is another invasive diagnostic procedure used in clinical settings. It involves obtaining a small tissue sample from the patient’s body, which is then examined in the laboratory, typically through histological analysis, to confirm the condition of the pancreas.

In the context of PC, tissue biopsy involves the procedure of acquiring a sample of pancreatic tissue from a patient intended for subsequent analysis, including histological examination and immunohistochemistry. For diagnostic purposes, various tissue biopsy methodologies are available as alternatives to the conventional open surgery approach. These include laparoscopic procedures, FNA, and FNB aimed at mitigating the potential of complications, minimizing the postoperative impact, and reducing recovery time.

Laparoscopy, used for diagnostic purposes, involves the insertion of a camera (laparoscope) through small incisions in the abdomen. This approach allows for the localization of the pancreatic tumor, aiding in cancer staging and guiding the resection decision [97]. In contrast to invasive laparoscopy, FNA and FNB are minimally invasive methodologies that entail endoscopic procedures inserted through the oral cavity to the surrounding pancreatic region. The aforementioned EUS-FNA and endoscopic ultrasound fine-needle biopsy (EUS-FNB), utilizing a larger needle, employ an endoscope equipped with an ultrasound probe to visualize the precise location of suspected lesions and to extract biopsy samples or tissue for further examination [82].

Concerning potentially suspicious lesions, histological examination serves as a straightforward approach to analyzing tissue types under a microscope, involving processes such as fixation, sectioning, and staining. Conventional histological staining entails retaining the chemical compounds of the tissue to differentiate various protein types, while immunohistochemistry (IHC) employs antibodies labeled with colored dyes to localize specific proteins within tissues or cells [98].

IHC is the definitive method for diagnosing PC [99]. IHC is favored due to its ease of application, availability, cost-effectiveness, and ability to detect PC in its early stages [100]. One of the commonly used IHC biomarkers is carbohydrate antigen 19-9 (CA19-9), which has been approved by the United States Food and Drug Administration (FDA) for use as a PC marker [101]. The intensity of CA19-9 staining in fixed paraffin tissue can indicate the presence of PC as low-intensity values correspond to a low level of sialyl Lewis antigen—CA19-9 binding [102]. Other IHC biomarkers, such as Galectin-1, Maspin, KOC, S100P, and pVHL, also exhibit high sensitivity and specificity in the PC detection [99,103]. Additionally, CA19-9 biomarkers have been adapted for the plasma-based metabolite analysis [104]. However, several drawbacks exist in the utilization of IHC biomarkers as indicators for PC, such as invasive method, the lack of a standardized system for immunostaining scoring, disparities in the expression of IHC markers across samples, and difficulties in obtaining appropriate matches between cases and controls, which may undermine the precision of IHC biomarkers [99].

### 3.4. Biomarker Detection in Bodily Fluids

Conversely, molecular tumor profiling can serve as a cancer diagnosis tool. Still, the current approach of profiling tumors by invasive tumor sampling and histological analysis faces challenges due to the heterogeneous nature of resected tumor tissue samples [105,106]. This limits the quantity and quality of collected samples. Consequently, non-invasive techniques have gained popularity for their convenience in enabling repeated sampling throughout treatment. Although imaging techniques can detect PC, their effectiveness at early stages is limited by the small size of cancer tumors [107]. In contrast, precision medicine relies on a comprehensive understanding of an individual’s health status and disease stage and uses omics, such as genomics, transcriptomics, metabolomics, glycomics, proteomics, and volatolomics, to identify cancerous markers and improve cancer screening, diagnosis, and treatment [108].

LB is a promising diagnostic method with great potential in cancer detection by assessing a wide range of bodily fluids, including blood, urine, saliva, feces, and cerebrospinal fluid. It utilizes various analytes, such as circulating tumor cells (CTCs), circulating tumor DNA (ctDNA), tumor protein, glycoprotein, platelets, tumor-derived extracellular vesicles (microvesicles, exosomes), microRNA (miRNA), messenger RNA (mRNA), and volatile organic compounds (VOCs), to identify specific biomarkers with high sensitivity. These analytes provide valuable information for tumor detection, including mutational profiles, epigenetic changes, nucleotide deletion, mutational deletions, translocations, gene rearrangement, copy number alterations, tumor heterogeneity, tumor subtypes, glycomic profile, volatolomic profile, protein expression, and other biomarkers associated with LB for cancer diagnosis [105,109]. However, it is essential to note that each analyte may contain only some of the biological information needed. For example, serum can detect proteins, glycolipids, adhesion molecules, and epigenetic codes, but it may not detect VOCs.

The selection of appropriate analytes for a specific diagnostic application requires careful consideration to ensure that the desired biological information is captured effectively. Consequently, there are ongoing efforts to develop challenging technologies for cancer screening purposes, particularly in large populations. Early tumor detection through LB is of utmost importance as it enables the identification of prevalent subtle changes in metabolism, marker secretion, DNA levels, or cellular characteristics, even in individuals who may not exhibit evident symptoms. This approach provides an opportunity for heightened awareness and early intervention, resulting in improved chances of full recovery and reduced healthcare costs.

LB techniques offer a less invasive or non-invasive means of sampling bodily specimens, and they can be easily performed by individuals themselves or healthcare providers. This lessens the stress associated with traditional diagnostic procedures and optimizes the utilization of professional expertise.

#### 3.4.1. Protein and Glycan Markers

Presently, CA19-9 is the predominant biomarker for diagnosing PC, which can be utilized in various methods such as histological analysis through tissue samples [92], cell surface marker via serum, or serum CA19-9 levels [110]. The sensitivity and specificity of CA19-9 are acceptable, with values of 75% and 85%, respectively, and an area under the curve (AUC) of 0.84 [111]. Nevertheless, the drawback of using CA19-9 lies in its lack of specificity, as false positive results in the presence of other bodily cancerous parts such as the colon, gastric, and biliary tract, and it fails to differentiate between PC and chronic pancreatitis [99,112,113]. Therefore, using CA19-9 alone for screening is insufficient, and a combination of CA19-9 with other biomarkers can enhance the screening and diagnostic efficacy by increasing sensitivity and specificity [111,114,115,116,117]. To this end, several reports have suggested combining CA19-9 with other biomarkers, such as Carcinoembryonic antigen (CEA). The use of CEA as a biomarker for PC has been increasing, with studies reporting that the sensitivity and specificity range from 64–95% and 50–92%, respectively. Despite the moderate performance values of CA19-9 and CEA, different cutoff values may influence the diagnostic value. Nonetheless, several studies still recommend using these biomarkers for early PC detection due to their convenience, efficiency, and non-invasiveness [111].

Furthermore, the IMMray^®^ PanCan-d test, developed by Immunovia, Inc. in Marlborough, MA, USA, employs a panel of nine serum biomarkers, including CA19-9, to detect early-stage PDAC from blood samples. A total of 586 individuals, comprising 167 PDAC patients (including 56 patients at stages I and II), 203 individuals at high risk of PDAC, and 216 healthy controls, were included in the study. The test demonstrated a sensitivity of 92% for all stages of PDAC and 89% for stages I and II, along with a specificity of 99% [118]. In addition, other urinary biomarkers, including polyamines, have shown high sensitivity in distinguishing PC from chronic pancreatitis. Salivary biomarkers have also been studied, with N^1^, and N^12^-diacetylspermine from saliva collected from patients with PC showing high AUC values [114]. However, the protein markers in saliva have a limitation in that false positives can occur in patients with other types of cancer. Adnab-9, a murine monoclonal antibody, has been employed in fecal research to detect PC, achieving a sensitivity of 80% and specificity of 87% [119,120].

Additionally, protein glycosylation is a prevalent posttranslational modification that significantly impacts cellular processes, such as cell proliferation, differentiation, and intercellular communication [121]. Aberrant glycosylation is often associated with multiple diseases, including cancer, and can disrupt the functions and localizations of glycan-attached proteins. The abnormal glycosylation patterns found in cancer cells frequently involve the overexpression of truncated O-glycans and N-Acetyl-D-glucosamine-branching N-glycans, which can interfere with glycoprotein activity and signaling [122,123]. Recent studies employing a lectin microarray technique in PDAC tissues have detected an increase in sialylated glycans and bisecting N-acetylglucosamine, and fucosylation [124], which is consistent with the results obtained using mass spectrometry analysis in serum from PDAC patients, demonstrating a rise in α 2,6-linked sialylation, fucosylation of tri- and tetraanternary N-glycans. This independent validation showed 0.81 of AUC with sensitivity and specificity values of 0.75 and 0.72, respectively [125].

#### 3.4.2. Extracellular Vesicles

Extracellular vesicles (EVs) are spherical structures covered by a lipid bilayer that lack functional nuclei and replication capacity. These vesicles are shed from cell membranes and carry various molecules expressed by the originating cell, such as proteins, nucleic acids, and lipids, to mediate intercellular communication in the extracellular environment [126]. Based on their size, EVs are classified into three categories: exosomes (30–150 nm), micro-vesicles (50–100 nm), and apoptotic bodies (>1000 nm) [65]. In particular, small extracellular vesicles (sEVs) have been the focus of numerous studies, as they contain cancerous miRNA, mRNA, DNA, proteins, and other biomolecules derived from cancer cells [127,128]. Following their release into the extracellular environment, sEVs can induce normal gene expression and metabolism alterations, leading to tumor growth via angiogenesis, inflammation, and multi-pathway signaling upon binding and release into the host cells [129,130]. Early detection of PC can be achieved through the detection of EVs derived from PC, as these biomarkers can be detected from the initial stages of the disease [131].

PDAC patients are at an elevated risk of developing PC owing to the existence of EVs and an intratumoral microenvironment (stroma) [132]. The stroma primarily comprises immune cells and cancer-associated fibroblasts (CAFs) or pancreatic stellate cells communicating with tumor cells via EVs. CAFs-derived EVs contribute to carcinogenesis by transferring biomolecules, including the annexin A6/LDL receptor-related protein 1/thrombospondin complex found in EVs present in the serum of PC patients [133]. Additionally, miR-421 was found to be at high levels in EVs from PC specimens. PC is associated with epithelial-mesenchymal transition (EMT), which involves phenotypic changes in cells by enhancing the expression of mesenchymal factors through mediators such as LncRNAs [134,135]. A clinical study has shown that early-stage PC can be feasibly detected using protein biomarkers by purifying EVs from a blood test with a sensitivity of 71.2% at 99.5% specificity [136]. These suggest that the presence of these EVs in high-risk PDAC patients progressing to PC may potentially serve as a biomarker for the cancer [133].

#### 3.4.3. Circulating Tumor Cells and Circulating DNA

Circulating cell-free DNA (cfDNA) can be found in the bloodstream from various cells, including hematopoietic cells, healthy cells, apoptotic cells, and circulating tumor cells (CTCs) [137,138,139,140,141]. In the context of cancer diagnosis, the cfDNA method may not be suitable for clinical decision making due to other factors that can alter cfDNA levels, such as inflammation, exercise, smoking, trauma, and innate chromosomal abnormalities [142]. Within cfDNA, there is a specific subset known as circulating tumor DNA (ctDNA), which is derived from tumor cells. The ctDNA makes up less than 1% of cfDNA. The level of ctDNA released from cancer cells may vary depending on the sensitivity to chemotherapy, ranging from 0.1% to over 90% of all cfDNA Fields [143,144].

Detecting ctDNA levels indicates 48% of patients with localized cancer and >80% with late-stage pancreatic cancer. The ctDNA can potentially influence metastasis development or cancer gene expression in the body [145,146]. Galleri^®^ test, developed by GRAIL with headquarters in Menlo Park, CA, USA, is a multi-cancer detection (MCED) method that assesses cfDNA methylation in the blood sample. This test can evaluate a common cancer signal shared by over 50 cancer types. In PC’s case, the test demonstrates a sensitivity ranging from 63% to 100% across all stages, from stage I to stage IV [147]. Additionally, the Galleri^®^ test exhibits a high specificity of 99.5% in detecting the cancer signal across a wide range of more than 50 cancer types [148].

Another commercial product, known as CancerSEEK, developed by Thrive in Cambridge, MA, USA and owned by Exact Sciences in Madison, WI, USA, employs the MCED technique to identify cancer types by evaluating the concentrations of circulating proteins and mutation in cfDNA. The extensive clinical study involving 1005 patients and 812 healthy controls encompassed five types of cancer (ovary, liver, stomach, pancreas, and esophagus) and revealed a sensitivity range from 69% to 98% [149].

Detecting CTCs is crucial in understanding metastasis and tumor heterogeneity and is important for monitoring chemotherapy to avoid overtreatment and side effects [150,151]. Advanced techniques related to CTCs consist of specific immunoaffinity enrichment, size and density separation, and downstream multi-omics analysis [150]. Despite these advancements, obtaining CTCs of sufficient quantity and purity remains challenging due to factors such as their rapid degradation within a few hours and multiple tumor types within a blood sample [152].

Several immunoaffinity methods, including CellSearch^®^ (Manufactured by Menarini Silicon Biosystems in Huntingdon Valley, PA, USA), MACS^®^ (Manufactured by Miltenyi Biotec, with headquarters in Bergisch Gladbach, Germany), and Dynabeads^®^ (Manufactured by Thermo Fisher Scientific, with headquarters in Waltham, MA, USA), have been developed and are commercially available, with CellSearch^®^ being the FDA-approved method that uses magnetic beads coated with specific proteins CD45, cytokeratin, and epithelial cellular adhesion molecule (EpCAM) to remove leukocytes from a 7.5 mL blood sample [153,154]. In the context of patients with PC, CellSearch^®^ has a lower CTC detection rate with a sensitivity and specificity of 55.5% and 100%, respectively [155]. Another FDA-approved CTC isolation technique is ISET^®^ (Manufactured by Rarecells Diagnostics in Paris, France), which isolated CTCs based on size using an antibody-independent whole-blood filtration [156,157]. The investigators performed a study using ISET^®^ to predict PDAC by analyzing the cell of interest with a size greater than 8 μm in a 10 mL blood sample. Three types of CTCs, including epithelial CTCs, mesenchymal CTCs, and total CTCs, were linked with an increased risk of early recurrence. In the chemo-naïve group, mesenchymal CTCs demonstrated the highest predictive capacity with an AUC of 0.69, 52% sensitivity, and 82% specificity. In contrast, in the neoadjuvant group, the total CTCs had the highest predictive capacity, with an AUC of 0.75, 72% sensitivity, and 75% specificity [158]. Other methods for CTC isolation include OncoQuick^®^ (Manufactured by Greiner Bio-One, with headquarters in Kremsmünster, Austria), a density gradient separation technique, and multi-omics analysis involving RNA, DNA, and protein [156].

#### 3.4.4. Epigenetic Markers

Epigenetic modifications, particularly DNA methylation, involve the conversion of cytosine DNA bases into 5-methylcytosine (5mC) through DNA methylation. During this process, 5mC can further convert into 5-hydroxymethyl cytosine (5hmC) as an intermediate step when methyl groups are removed from the cytosine DNA [159].

In a specific study, patterns of 5hmC can be a biomarker for diagnosing cancer in several areas, including lung, liver, colon, gastric, and pancreas. Sets of genes (pancreatic genes: GATA4, GATA6, PROX1, ONECUT1, and MEIS2; cancerous genes: YAP1, TEAD1, PROX1, and IGF1) associated with 5hmC deregulation were identified in PDAC and found that activation of KRAS and inactivation of TP53 are associated with changes in 5hmC densities in PDAC tumors. The results from early-stage PDAC patients, 64 cancer patients, and 243 non-cancer patients showed an AUC of 0.92–0.94 in cfDNA samples and an AUC of 0.88 in tissue samples [160]. A different research study quantified the presence of 5hmC at specific DNA loci to identify cancer with a validation cohort comprised of 2150 individuals, of which 102 were PC patients and 2048 were non-cancerous patients. The model’s accuracy was confirmed using a separate set of independently processed samples that were blinded to the investigators, and it showed an early-stage sensitivity of 68.3% and a specificity of 96.9% [161].

Long non-coding RNAs (LncRNAs), which are RNA molecules longer than 200 nucleotides that do not encode proteins, function as competing endogenous RNAs (ceRNAs) with microRNAs (miRNAs) to modulate messenger RNA (mRNA) expression [162,163,164]. LncRNAs can compete with the endogenous target genes for miRNA binding. This can affect downstream mRNA expression levels and promote or inhibit tumor growth and cancer-related signaling pathways, including those involved in PC [165,166].

Recent studies have demonstrated the involvement of LncRNAs in various cellular processes, including the regulation of gene transcription, translation, and post-translational modification, highlighting their potential as biomarkers for detecting and prognosticating PC through the assessment of differential expression of specific LncRNAs between healthy and PC samples [162,167]. A comparison of the Long Intergenic Non-Protein Coding RNA, P53 Induced Transcript (LINC-PINT), and CA19-9 demonstrated that LINC-PINT, when measured in plasma, exhibits higher sensitivity (AUC = 0.87) than CA19-9 (AUC = 0.78) in detecting PC. The concentration of LINC-PINT in the plasma of PC patients was lower than that of healthy samples, indicating that LncRNAs in plasma may be used for early cancer detection [168].

#### 3.4.5. Volatile Markers

Volatile organic compounds (VOCs) refer to carbon-based molecules in a gaseous state at room temperature due to their high vapor pressure properties. VOCs, as the terminal cell metabolism with poor solubility in blood, traverse the circulatory system and are finally emitted in diverse biological fluids, such as blood, breath, feces, milk, saliva, semen, sweat, and urine [169,170]. In medical practices, these fluids are frequently used as VOC detection samples due to their easy accessibility and inexhaustibility [171]. However, collecting VOC samples requires specific procedures. For example, blood VOCs are obtained through a standard peripheral venipuncture [172], while breath VOCs are collected using a facemask–airbag system or breath sampler, which filters environmental air from the system [173].

VOCs in the bodily fluids can be used to analyze an individual’s metabolic state. Healthy individuals can contain up to 2746 VOCs in their body fluids [174], whereas cancer patients’ VOC profiles may result from the dysregulation of metabolic pathways associated with tumor growth [171]. Since readily accessible biomarkers are crucial in developing a personalized medicine strategy for therapeutics, extensive research has been conducted on VOCs as mostly non-invasive and sensitive biomarkers for clinical applications, focusing on an oncology [169,170]. Consequently, detecting disease-specific VOC profiles is a promising avenue for developing non-invasive diagnostic tests [170]. Several technologies, such as analytical-based, sensor-based, nanoparticle-based, and animal-based approaches, are employed to identify and quantify VOCs associated with PC.

##### Analytical-Based Technologies

Gas chromatography-mass spectrometry (GC-MS) is a method used for VOC analysis based on mass-to-charge ratios [175]. However, this method has high operational costs and the need for skilled technicians [176]. In a study, potential PC biomarkers, including glutamate, choline, 1,5-anhydro-D-glucitol, betaine, and methyl guanidine, showed high sensitivity values of 97.7% and 77.4%, specificity values of 83.1% and 75.8%, and AUC values of 0.943 and 0.83 for distinguishing early-stage PC patients from healthy individuals in two distinct cohorts [177].

Selected ion flow tube mass spectrometry (SIFT-MS) offers a simpler alternative for VOC detection using chemical ionization and MS. In bile analysis, SIFT-MA differentiated 65 PC cases from 23 chronic pancreatitis cases with a high sensitivity of 93.5%, specificity of 100%, and AUC of 0.98 using VOC levels of ammonia, acetonitrile, and trimethylamine [178]. Another study examined VOCs in urine to distinguish between 15 cases of cholangiocarcinoma and PC and 29 cases of benign biliary groups, including chronic pancreatitis and papillary stenosis. The model utilizing VOC levels of 2-propanal, carbon disulfide, and trimethylamine achieved a sensitivity of 93.3%, specificity of 61.5%, and AUC of 0.83, respectively [179].

FAIMS hold potential in detecting VOCs related to cancer, although its popularity is not as widespread as GC-MS or SIFT-MS [180]. In one study aiming to discriminate between 69 cases of PC patients and 52 healthy controls using FAIMS with urine samples, the obtained sensitivity and specificity were 79%. These findings indicate the need for improvements in the measurement system to enhance the performance of FAIMS in VOC analysis for PC detection.

##### Sensor-Based and Nanoparticle-Based Techniques

Analytical measurements offer cancer analysis, but they have limitations in terms of scalability, cost, and time consumption [181]. Recent advancements include electronic nose (e-nose) detectors capable of identifying complex VOC mixtures without pinpointing individual compounds. E-nose mimics mammalian olfactory systems, utilizing multiple sensors analyzing VOC responses at low concentrations [182,183]. These response patterns can be analyzed using artificial neural networks with pattern recognition algorithms and a library database [184]. E-nose technology can distinguish between healthy samples and cancer through exhaled breath analysis at the lowest limit of detection (LOD), typically in the range of parts-per-billion (ppb) to parts-per-million (ppm) concentration [185]. Thus, ensuring the proper breath VOC sample collection is essential to prevent degradation. The e-nose device typically consists of three major components: a sample delivery system responsible for the collection, extraction, and introduction of VOCs to a detection system and a data computing system [181]. There are various sensor materials, following the examples below:

Metal-oxide semiconductor (MOS) sensors are popular due to cost-effectiveness, but they require high operating temperatures or humidity to initiate surface reactions and have limited sensitivity to certain VOC fields [185,186]. Notable examples include SprioNose^®^ (manufactured by Breathomix BV in Leiden, The Netherlands) and Aeonose^TM^ (manufactured/ developed by The eNOSE company in Zutphen, The Netherlands).
SprioNOSE^®^ enables real-time measurement of VOCs and connects to the online BreathBase platform. In a clinical study on early-stage lung cancer, it demonstrated a sensitivity of 86%, specificity of 89%, and AUC of 0.90 [187].Aeonose^TM^ has conducted clinical studies focusing on lung cancer, resulting in a sensitivity of 83% and specificity of 84% when comparing 52 lung cancer patients with a benign group of 93 individuals [188]. In another study involving PDAC (*n* = 29), chronic pancreatitis (*n* = 56), and non-cancer control (*n* = 74), distinct VOC profiles were observed among the three groups. Discrimination between PDAC and chronic pancreatitis resulted in a sensitivity of 83%, specificity of 71%, accuracy of 0.75, and AUC of 0.83. Differentiating the non-cancer group from PDAC resulted in a sensitivity of 83%, specificity of 82%, accuracy of 0.83, and AUC 0f 0.87 [189].


Conducting polymer gas sensors detect changes in electrical resistance caused by gas adsorption on the sensor surface [190]. They offer high sensitivity, rapid response time, ease of synthesis, and the ability to operate under ambient conditions [191].
Cyranose^®^ 320 (manufactured by Sensigent in Baldwin Park, CA, USA) is a commercial odor-monitoring device used in medical research, including studies on diabetes and cancer. For instance, in a study comparing a non-cancer group (consisting of smokers and non-smokers, *n* = 223) with a lung cancer group (*n* = 252), reported sensitivities ranged from 95.8% to 96.2%, and specificities ranged from 90.6% to 92.3% [192]. Another example involved an early-stage lung cancer group, with a non-cancer group of 188 individuals and a lung cancer group of 56 individuals. This study showed sensitivity, specificity, and accuracy of 83%, 86%, and 85%, respectively [193]. These sensors have been investigated for other types of cancer, such as bladder cancer (sensitivity of 93.3% and specificity of 86.7% in a non-cancer group, *n* = 30, and a cancer group, *n* = 30) [194], and colorectal cancer (sensitivity of 62–85% and specificity of 86–87% in a non-cancer group, including advanced adenoma and healthy control, *n* = 117, and a cancer group, *n* = 40) [195].


Acoustic wave gas sensors, including bulk acoustic wave (BAW) and surface acoustic wave (SAW) mechanisms, detect VOC profiles through changes in mechanical waves. However, a piezoelectric acoustic wave device (known as quartz microbalance (QMB)) is for mass accumulation detection [196]. Carbon nanotubes (CNTs) have been integrated into e-nose systems to quantitatively monitor solubility, polarity, and chemical associations [197,198].

However, e-nose systems face challenges detecting low biomolecule concentrations and dealing with noise and environmental factors such as humidity and temperature, potentially causing inaccurate detection. To improve accuracy, the e-nose industry needs robust data analysis techniques, such as artificial neural networks [181].

##### Animal-Based Techniques

Cancer diagnosis requires timely detection for effective treatment. While e-nose sensors aim to mimic the animal olfactory system and have limitations in odor selectivity and development time. Actual animal sensors have an advantage due to their natural olfactory networks, capable of discerning various metabolic profiles of normal and abnormal cells thereby yielding different metabolites [199]. Living organisms, particularly dogs known for their remarkable olfactory acuity, have been used to detect diseases, including cancer [200].

Previous research has shown that training dogs for cancer screening purposes is time-intensive, spanning up to 16 months to ensure the reliability and accuracy of their olfactory abilities. However, not all dogs successfully complete the training and testing phases, leading to time inefficiencies and a limited number of qualified dogs for deployment in such applications [201]. However, qualified dogs could sniff the cancer samples with high accuracy. A commercial dog sniff test known as the Cancer Scent Screening Kit (manufactured/developed by BioScent DX, Inc. in Myakka City, FL, USA) has reported that four dogs (excluding one unmotivated dog) were able to distinguish blood serum from a patient with lung cancer and healthy control with sensitivity of 96.6% and specificity of 97.5% [202]. Overall, utilizing sniffer dogs for cancer detection through scent recognition represents an alternative option. The detection accuracy may vary based on individual biological factors and the intricacies of the training process.

While the olfactory system in animals generally functions effectively, there are differences among species in their ability to detect cancer. For instance, dogs require training to detect cancer, whereas nematodes exhibit innate behavioral responses to cancer urine. These differences can be attributed to the complexity of their olfactory sensory systems and the genetic makeup of their olfactory receptor genes. The olfactory pathway involves specialized chemosensory cells called olfactory sensory neurons (OSNs) in the nose, which detect and transmit odorant signals. These signals then undergo peri-receptor processes, ultimately guiding behavior and forming olfactory memories [203]. Each animal has a unique ability to respond to different concentrations of VOCs or specific ratios of odorant mixtures in the surrounding environment through olfactory receptors (ORs) [204]. Many of these receptors belong to the family of G protein-coupled receptors (GPCRs), characterized by their seven transmembrane domain proteins. When a ligand (odorant) binds to the ORs, it initiates a cascade of events within the animal’s olfactory system, leading to the generation of olfactory sensations and subsequent behavioral or physiological responses [203].

*Caenorhabditis elegans* (*C. elegans*), a species of nematode, has emerged as a prominent model organism for the investigation of cancer odor detection systems due to its favorable attributes, including its natural habitat to compounds, its small size, short life cycle (3 days at 20 °C), high reproductive capacity (~1000 eggs), and the absence of ethical considerations [205,206]. The *C. elegans* nervous system contains 32 presumed chemosensory neurons directly or indirectly exposed to the environment [207].

Approximately 1300 genes and 400 pseudogenes encoding chemosensory GPCRs enable nematodes to possess a wide range of receptors capable of recognizing diverse odor molecules. Notably, chemosensory genes constitute the largest gene family in *C. elegans*, accounting for approximately 8.5% of the entire genome. This unique genetic makeup, coupled with the presence of multiple chemosensory neurons, has led to *C. elegans* being considered an ideal model organism for screening various compounds, including those relevant to cancer detection. Remarkably, each ORN in *C. elegans* demonstrates an exceptional capability to detect a broader spectrum of odorants compared to their mammalian counterparts [207,208]. Consequently, identical GPCRs can be expressed on different cells, and the same odorant or ligand can bind to different GPCRs located on functionally diverse neurons, depending on their concentration [207].

*C. elegans* displays responses to a wide range of VOCs and soluble mixtures, including alcohols, ketones, aldehydes, esters, amines, sulfhydryl, organic acids, and aromatic and heterocyclic compounds. Water-soluble compounds can be detected in the nanomolar range, while salts and amino acids are detected in the micro to the millimolar range. *C. elegans* exhibits an innate chemotactic response toward specific samples, such as urine, through its amphid sensory neurons, specifically AWA, AWB, and AWC. The AWA and AWC sensory neurons demonstrate an attractive behavior towards chemical compounds such as benzaldehyde, butanone, isoamyl alcohol, 2,3-pentanedione, diacetyl, pyrazine, and 2,4,5-trimethylthiazole. In contrast, AWB neurons are associated with a repulsive response to odors such as 2-nanonone. The involvement of chemosensory neurons can be confirmed using calcium imaging to ensure consistency with the results obtained from the chemotaxis assay [206,209].

The chemotaxis assay investigates the attractive and repulsive components of *C. elegans*’ response to stimuli [206,210]. The procedure involves dividing a petri dish into four quadrants, with two left quadrants containing urine spots and the other two right quadrants without urine. Subsequently, *C. elegans* are placed in the center of the plate and allowed to move freely for 30–60 min, after which the nematodes are scored. The chemotaxis index is calculated by subtracting the number of nematodes in the no-urine region (b) from the number of nematodes in the urine (1) and then dividing this difference (a − b) by the total number of nematodes (a + b). The resulting chemotaxis index can range from −1 to 1, where values between −1 and 0 indicate a negative index, representing repulsion from the sample, and values between 0 and 1 indicate a positive index, indicating attraction towards the sample (illustrated in Figure 4). The Nematode-nose (N-NOSE) cancer screening test uses a cut-off value of zero for the boundary of the positive–negative index. If the N-NOSE index is greater than zero, it suggests a likelihood of cancer, while a value lower than zero indicates the absence of a cancer [211].

N-NOSE, manufactured/ developed by Hirotsu Bio Science in Tokyo, Japan, represents a pioneering approach to MCED test designed for early cancer detection. This innovative method combines the utilization of *C. elegans* with the chemotaxis assay to observe the nematodes’ behavior of being attracted to urine samples from cancer patients and repelled by urine samples from healthy individuals. Urine samples are diluted in a 10-fold to 100-fold range for optimal results. The first generation of N-NOSE serves as a primary multi-cancer screening tool capable of detecting the presence of 15 cancer types: stomach, colorectal, lung, breast, pancreatic, liver, prostate, uterine, esophageal, gall bladder, bile duct, kidney, bladder, ovarian, and oral/pharyngeal cancers. However, it is important to note that this method cannot specify the specific cancer types or provide information regarding the staging of the disease [211].

In a clinical study conducted by Inaba et al., urine samples from 32 cancer patients at all stages and 143 control individuals were examined. The study reported an AUC of 0.92 and 0.90 with a 10-fold and 100-fold dilution, respectively. Additionally, when both dilution methods were combined, the study demonstrated a high sensitivity of 87.5% and a high specificity of 90.2%, further confirming the reliability of N-NOSE as a diagnostic tool [212]. Moreover, another clinical study focusing on early-stage gastrointestinal cancer, including colorectal (*n* = 67), gastric (*n* = 58), pancreatic (*n* = 24), biliary tract (*n* = 9), esophageal (*n* = 18), and gallbladder cancer (*n* = 4) evaluated the efficacy of N-NOSE. The study revealed that N-NOSE exhibited an AUC greater than 0.80 at all stages, including the early stage, and for all types of gastrointestinal cancer. In comparison, commonly used biochemical markers such as CEA and CA19-9 showed lower AUC values, with CEA below 0.75 and CA19-9 below 0.69. Notable, for PC detection, N-NOSE achieved an AUC of 0.862, while CEA and CA19-9 yielded AUC values of 0.782 and 0.924, respectively. These findings suggest that combining N-NOSE with CA19-9 could enhance the accuracy and effectiveness of N-NOSE in early cancer detection of PC [213]. In a study conducted at Saitama Medical University involving 104 PC patients and 95 control individuals, N-NOSE demonstrated significantly higher sensitivity in detecting early-stage PC compared to the traditional biomarkers CEA and CA19-9 [214]. Similarly, a pilot study conducted at Osaka University analyzed urine samples from 83 patients with PC at various stages using N-NOSE. The test exhibited strong performance, with AUC values of 0.845 and 0.820 at 10-fold and 100-fold dilutions, respectively. Significant differences were observed in the *p*-values obtained before and after surgical removal of PDAC. Furthermore, in a subsequent blind study using urine samples from 27 individuals (11 early PC cases and 17 healthy samples), the chemotaxis index in early PC cases was significantly higher than in the healthy samples at both dilutions [215]. These findings from all these studies indicate the potential of N-NOSE as a valuable tool for the early detection and monitoring of PC. Importantly, N-NOSE is non-invasive since it relies on urine samples.

## 4. Market of PC Screening Test

There are various modalities available for the detection of PC, including both imaging-based and LB techniques. Imaging modalities such as ultrasound, CT scan, and EUS provide an initial assessment, but their specificity depends on the depth of information, and they can be invasive and expensive. On the other hand, LB techniques, which involve collecting tissue, blood, or urine, offer a potentially more cost-effective alternative. Traditional protein markers such as CA19-9 and CEA can be detected using commercial products such as IMMray^®^ PanCan-d test, Galleri^®^ test, and CellSearch^®^ test, with costs ranging from USD 900 to USD 995 [216,217,218] in Table 1.

In contrast to protein markers, volatile markers are detected using GC-MS, which typically costs less than USD 75 per sample. However, access to this method may require approval from healthcare providers. E-nose technology, utilizing commercial sensors such as SprioNose^®^, AeonoseTM, and Cyranose^®^320, has been developed for cancer detection, primarily targeting lung cancer rather than PC. These devices can be costly, ranging from USD 9000 to USD 150,000 [219,220].

Animal-based methods, such as using trained dogs, are challenging due to the need for extensive training and the unavailability of PCs in the current database. However, the test kits for this method are relatively inexpensive at USD 50 [202]. In Japan, the N-NOSE multi-cancer screening test, which utilizes nematodes, can detect various cancer types, including PC, at a reasonable price of USD 109 [211]. The N-NOSE Plus, specifically for PCs (USD 505), has been launched in the Japanese market, offering a high AUC of 0.865 and a more affordable option than other commercial products currently available [221]. It is important to note that N-NOSE serves as a primary-secondary screening test, and N-NOSE Plus Pancreas allows for specific pancreatic cancer presence assessment.

**Table 1 biomedicines-11-02557-t001:** Cancer detection tests.

Detection Modalities	Instrument, Technique	Commercial Name, Company	Available Countries	Cancer Type	Sample	Sensitivity	Specificity	AUC	Accuracy	Cost	Ref.
Image modalities	Ultrasound			PC	Tissue	-	-	-	50–70% ^†^	USD1355 ^*^	[78] ^†^ [222] ^*^
	CT scan	-		PC	Tissue	75–100% ^†^	70–100% ^†^	-	73% ^†^	USD325.60–3394.00 ^*^	[90,91] ^†^ [223] ^*^
	PET/CT			PC	Tissue	46–71% ^†^	63–100% ^†^	-	-	USD1000 ^*^	[89,92] ^†^ [211] ^*^
	MRI			PC	Tissue	81–99% ^†^	70–93% ^†^	-	-	MRI only USD587.92–3869.00 ^*^	[83,96] ^†^ [223] ^*^
	MRCP			PC	Tissue	85–87% ^†^	93–95% ^†^	-	-	MRCP only USD71.45–USD787.00 ^*^ MRI + MRCP USD659.37–4656.00 ^*^	[88] ^†^ [223] ^*^
	CE-EUS			PC	Tissue	91% ^†^	93% ^†^	-	-	EUS USD307.23–5370.0 ^*^	[84,85] ^†^ [223] ^*^
	EUS-FNA			PC	Tissue	95–97% ^†^	67–76% ^†^	-	-	USD1,681 ^*^	[74,79,80,81] ^†^ [216] ^*^
Protein markers	CA19-9			PC	Blood serum	75% ^†^	85% ^†^	0.84 ^†^		-	[111] ^†^
	CA19-9	IMMray^®^ PanCan-dtest, Immunovia, Inc.	US	PC	Blood serum	92% ^†^	99% ^†^	-		USD995 ^*^	[118] ^†^ [224] ^*^
	CEA Antibody			PC	Blood serum	64–95% ^†^	50–92% ^†^	-		-	[111] ^†^
					Feces	80% ^†^	87% ^†^	-		-	[119,120] ^†^
Glycan markers	Lectin microarray			PC	Blood serum	75% ^†^	72% ^†^	0.81 ^†^	-	-	[125] ^†^
Exosomes	EVs			PC (Early-stage)	Blood serum	71.2% ^†^	99.5% ^†^	-	-	USD350 ^*^	[136] ^†^ [217] ^*^
	cfDNA	Galleri^®^ test, GRAIL	US	PC	Blood serum	63–100% ^†^	99.5% ^†^	-	-	USD949 ^*^	[147,148] ^†^ [218] ^*^
		CancerSEEK test, Thrive	US	PC and others (Ovary, Liver, Stomach, and Esophagus)	Blood serum	69–98% ^†^	-	-	-	<USD500 ^†^	[149] ^†^
	CTCs	CellSearch^®^	US	PC	Blood serum	55.5% ^†^	100% ^†^	-		USD900 ^*^	[155] ^†^ [225] ^*^
		ISET^®^	France	PC	Blood serum	52–72% ^†^	75–82% ^†^	0.69–0.75 ^†^		-	[158] ^†^
Epigenetic markers	5mC			-	Blood serum	68.3% ^††^	96.9%^††^	0.92–0.94 ^†^		-	[160] ^†^ [161] ^††^
	LncRNAs			PC	Plasma metabolites	-	-	0.87 ^†^	-	-	[168] ^†^
Volatile markers	GC-MS			PC	Plasma metabolites	77.4–97.7% ^†^	75.8–83.1% ^†^	0.83–0.943 ^†^	-	USD25–USD75 ^*^	[177] ^†^ [219] ^*^
	SIFT-MS			PC	Urine	93.3–93.5% ^†^	61.5–100% ^†^	0.83–0.98 ^†^	- -	-	[178,179] ^†^
	FAIMS			PC	Urine	79% ^†^	79% ^†^	-	-	-	[180] ^†^
	e-nose MOS sensor	SprioNose^®^, Breathomix BV	The Netherlands	Lung	Breath	86% ^†^	89% ^†^	0.90 ^†^	-	SprioNose^®^ and Aeonose^TM^ USD11,000–USD150,000 ^*^	[187,188] ^†^ [220] ^*^
		Aeonose^TM^, eNOSE	The Netherlands	Lung	Breath	83% ^†^	84% ^†^	-	-		[188] ^†^
				PC	Breath	83% ^†^	71–82% ^†^	0.83–0.87 ^†^	75–83% ^†^		[189] ^†^
	Conducting polymer gas sensor	Cyranose^®^320, Sensigent	US	Lung	Breath	95.8–96.2% ^†^	90.6–92.3% ^†^	-	-	USD$9000–$11,500 ^*^	[192] ^†^ [226] ^*^
				Lung (Early-stage)	Breath	83% ^†^	86% ^†^	-	85% ^†^		[193] ^†^
				Bladder	Breath	93.3% ^†^	86.7% ^†^	-	-		[194] ^†^
				Colorectal	Breath	62–85% ^†^	86–87% ^†^	-	-		[195] ^†^
	Animal-based -Dogs	Cancer Scent Screening Kit, BioScent DX, Inc.	US	Lung	Blood	96.7% ^†^	97.5% ^†^	-	-	USD50.00 ^*^	[202] ^†^ [227] ^*^
	*-C. elegans*	N-NOSE, Hirotsu Bio Science, Inc. (Tokyo, Japan)	Japan	PC	Urine	87.5% ^†^	90.2% ^†^	0.90–0.92 ^†^ 0.862 ^††^ 0.820–0.845 ^†††^	-	N-NOSE USD109 ^*^ N-NOSE plus pancreas USD505 ^**^	[211] ^*^ [212] ^†^ [213] ^††^ [215] ^†††^ [228] ^**^

^*^,^**^ indicate sensitivity, specificity, AUC, and accuracy referred from the respective references. ^†^, ^††^, ^†††^ indicate a different set of prices referred from their respective references.

## 5. Conclusions

In conclusion, PC is a challenging disease with a high mortality rate, and early detection is crucial for improving prognosis and patient outcomes. While various diagnostic methods exist, the N-NOSE multi-cancer screening test stands out as the best option. This test utilizes nematodes and has been launched in the Japanese market at an affordable price. The N-NOSE plus specifically targets PC, offering a high AUC. Compared to other commercial products, N-NOSE provides a more cost-effective solution for early detection. It is important to note that N-NOSE served as a primary-secondary screening test, and additional diagnostic tests are necessary for confirmation.

The market for PC screening tests includes imaging-based techniques and LB methods. While imaging modalities such as ultrasound, CT scan, and EUS provide initial assessments, they can be invasive and expensive. LB techniques, which involve collecting samples from tissue, blood, or urine, offer a potentially more cost-effective alternative. Among the LB methods, the N-NOSE test is a promising option due to its affordability, high AUC, and ability to detect various cancer types and biomarkers. Developing more precise and non-invasive screening methods and standardized scoring systems is crucial for enhancing PC detection and improving patient outcomes.

## Figures and Tables

**Figure 1 biomedicines-11-02557-f001:**
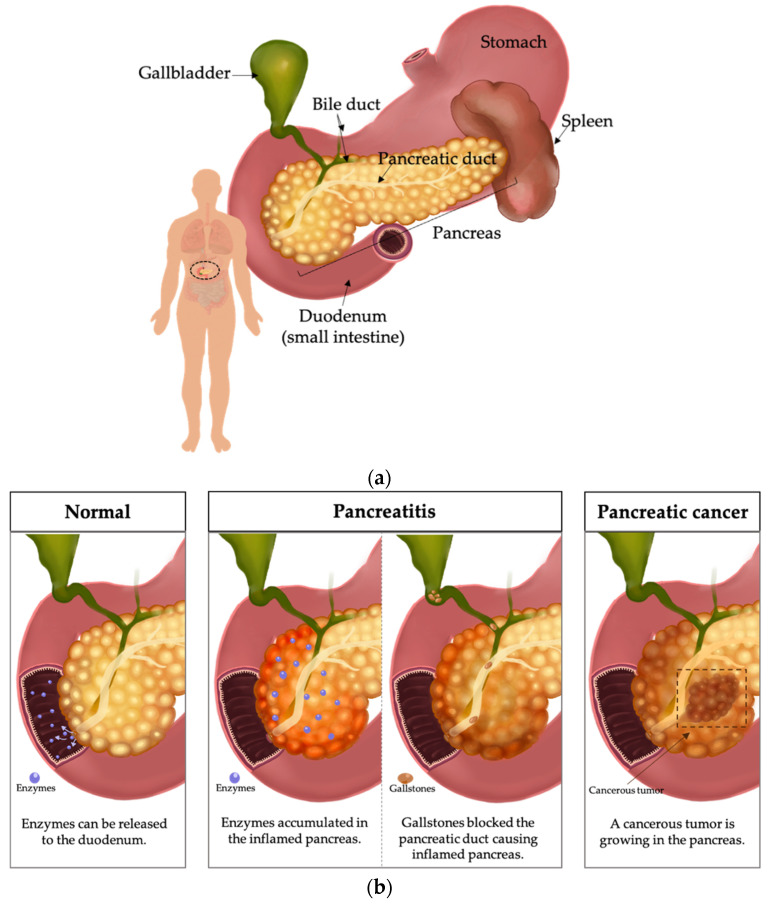
Illustration of the pancreas. (**a**) Location of the pancreas in the human body, (**b**) comparison among the normal pancreas, pancreatitis, and pancreatic cancer.

**Figure 2 biomedicines-11-02557-f002:**
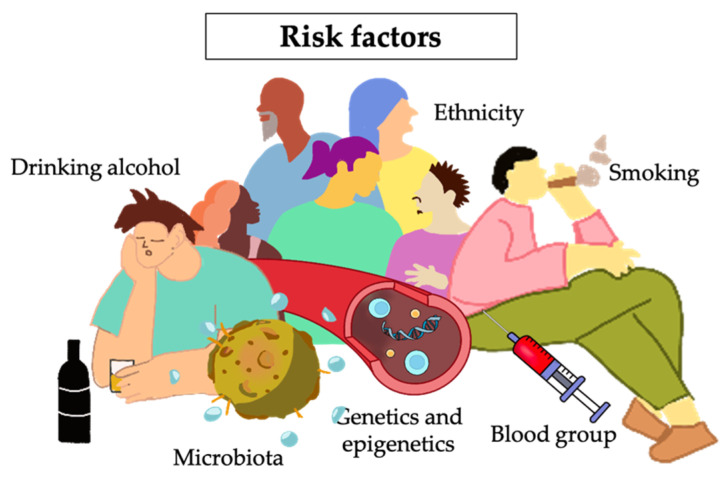
Risk factors of PC.

**Figure 3 biomedicines-11-02557-f003:**
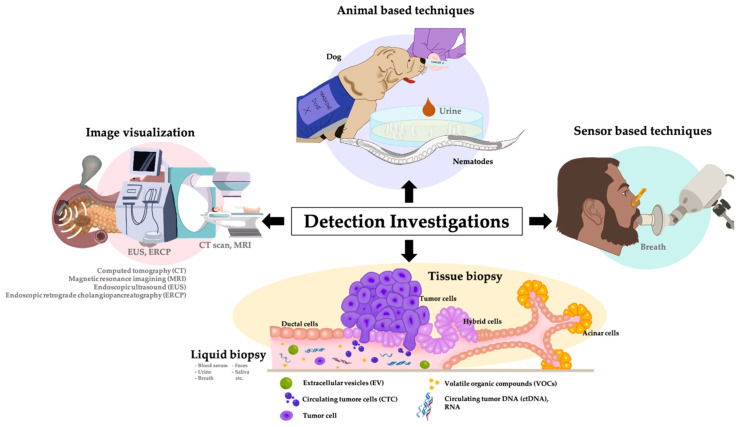
Detection investigations include (1) image visualizations, (2) tissue biopsy and liquid biopsy, (3) sensor-based techniques, and (4) animal-based techniques.

**Figure 4 biomedicines-11-02557-f004:**
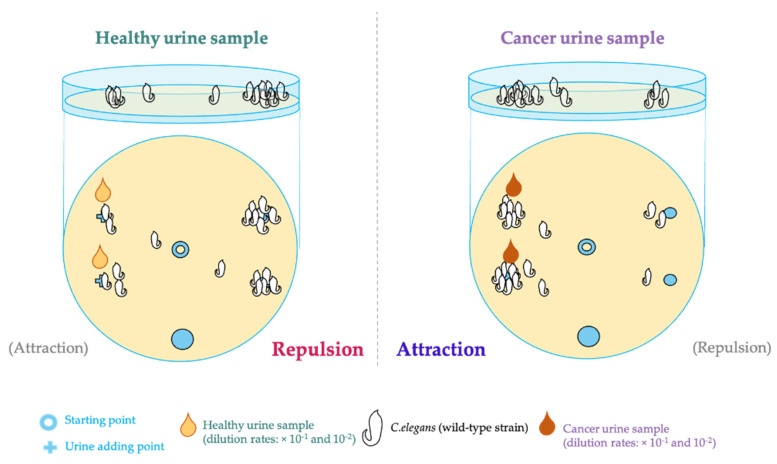
Chemotaxis assay of N-NOSE. In the chemotaxis assay, *C. elegans* exhibits repulsion towards healthy urine samples, while it shows attraction towards cancer urine samples.

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
