# Peer review of "Pancreatic Cancer and Detection Methods"

_biomedicines, 2023, doi:10.3390/biomedicines11092557_

Round 1

Reviewer 1 Report

Umbhorn Ungkulpasvich and coauthors propose an interesting workflow to detect pancreatic cancer. In this review authors nicely explained pancreatic cancer development and their detection methods. Despite the promise, the manuscript has several important flaws that need to be addressed before publication:

1.    2nd Line of the abstract needs to be re-written. It breaks the flow.

2.    Is detection a keyword?

3.    Full name of PRSS1 gene.

4.    Need to elaborate on all types of PC in section 2.1.

5.    Add space before ref 25 in the text.

6.    Add the % increasing risk caused by alcohol.

7.    Describe histological analysis in more detail.

8.    Explain a little bit about ERCP also.

9.    Full form of EUS-FNA

10. Why the author added, “As a result, CT or MRI scans are often used for primary evaluation instead.” In the Ultrasound area.

11. Write some lines for only CT scan.

12. Why author add the last line about tissue biopsy in the MRI area, they already explained tissue biopsy in the next section.

13. Explain about tissue biopsy process, not immunohistochemistry.

14. The author describes Extracellular vesicles, but the heading is exosomes.

15. 130 and 131 references are not in the proper area.

16. Why do authors elaborate more about Volatile markers?

Author Response

Point-by-Point Responses to the Comments Raised by the Reviewers

All reviewers:
We would like to convey our gratitude to all reviewers for their perceptive feedback on our manuscript. Their remarks have played a pivotal role in enhancing the quality of the manuscript. We have meticulously addressed each of the raised points in a detailed manner. 
Highlighted in yellow are the sections of the manuscript that have been revised. In this letter, the references to page and line numbers pertain to the updated manuscript, including Figures and Tables. We are confident that our responses effectively attend to the concerns highlighted by the reviewers, and we are optimistic that the revised manuscript will meet the standards for publication.

Reviewer #1 comments to the Author
Umbhorn Ungkulpasvich and coauthors propose an interesting workflow to detect pancreatic cancer. In this review authors nicely explained pancreatic cancer development and their detection methods. Despite the promise, the manuscript has several important flaws that need to be addressed before publication:
Response to Reviewer #1’s comment: 
We value your feedback and the comprehensive assessment of our manuscript. Your insights and recommendations will undoubtedly contribute to the enhancement of our work. We are grateful for your constructive input.

Reviewer #1 comments 1:
1. 2nd Line of the abstract needs to be re-written. It breaks the flow.
Response to Reviewer #1’s comment 1:
We agree with the comment raised by reviewer #1. We have revised the second line of the abstract. Please refer to the revised manuscript (page 1, abstract lines 1-3).

Reviewer #1 comments 2:
2. Is detection a keyword?
Response to Reviewer #1’s comment 2:
We have amended the keyword “detection” to “detection methods”. Please refer to the revised manuscript (page 1, keywords line 1).

Reviewer #1 comments 3:
3. Full name of PRSS1 gene.
Response to Reviewer #1’s comment 3:
We have added the full name of PRSS1 gene. Please refer to the revised manuscript (page 3, line 83).

Reviewer #1 comments 4:
4. Need to elaborate on all types of PC in section 2.1.
Response to Reviewer #1’s comment 4:
We agree with the comment raised by reviewer #1. In response, we have revised the section 2.1, “Incidence of Pancreatic Cancer”, in order to present a more comprehensive explanation of all pancreatic cancer types. We have outlined that …
“Within the realm of PC, malignancies can arise from either exocrine and endocrine cells. The most prevalent form of PC originates from exocrine cells, specifically known as pancreatic ductal adenocarcinoma (PDAC). PDAC accounts for 90% of the total PC and it is a malignancy that has a dismal outcome, with a 5-year overall survival rate of less than 10%, even though the development of medical treatment [13]. This PC is caused by the abnormal growth of exocrine cells, accounting for over 90% of all pancreatic malignancies [14].
Among the exocrine cancers, a subtype is acinar cell carcinomas, characterized by their abnormal secretion of enzymes. Furthermore, there are fewer common types of exocrine PC, such as adenosquamous carcinomas, squamous cell carcinomas, signet ring cell carcinomas, undifferentiated carcinomas, and undifferentiated carcinomas with giant cells. However, these subtypes are much less common and account for a smaller percentage of PC cases [15]. Conversely, the remaining PC subtypes develop improperly from endocrine or neuroendocrine cells, a phenomenon intricately linked to hormone production [16].”
 Please refer to the revised manuscript (page 4, lines 115 - 129).

Reviewer #1 comments 5:
5. Add space before ref 25 in the text.
Response to Reviewer #1’s comment 5:
We have added space before reference 25 within the sentence. Please refer to the revised manuscript (page 5, line 173).

Reviewer #1 comments 6:
6. Add the % increasing risk caused by alcohol.
Response to Reviewer #1’s comment 6:
We have provided further clarification regarding the percentage increase in risk caused by alcohol following the statement below:
“The meta-analysis indicated a 20% elevated risk of PC among individuals who consumed 3 drinks per day (equivalent to 37.5 g of ethanol) when compared with non- or infrequent drinkers. In contrast, the relative risk did not increase significantly among light or moderate alcohol drinkers [51].”
Please refer to the revised manuscript (page 7, lines 232-236).

Reviewer #1 comments 7:
7. Describe histological analysis in more detail.
Response to Reviewer #1’s comment 7:
We have provided the supplementary information of histological analysis following the statement below:
“Concerning potentially suspicious lesions, histological examination serves as a straightforward approach to analyze tissue types under a microscope, involving processes such as fixation, sectioning, and staining. Conventional histological staining entails retaining the chemical compounds of the tissue to differentiate various protein types, while immunohistochemistry (IHC) employs antibodies labeled with colored dyes to localize specific protein within tissues or cells [92].”
 Please refer to the revised manuscript (page 10, lines 409-414).

Reviewer #1 comments 8:
8. Explain a little bit about ERCP also.
Response to Reviewer #1’s comment 8:
We have elucidated the ERCP information in the end of section 3.2.1 Ultrasound, because we want to connect the story of endoscopic procedure and X-ray technique before jump into the next section 3.2.2 CT.
This is the ERCP statement: “While the endoscopic procedure with the ultrasound probe is employed for tissue evaluation, ERCP integrates endoscopic techniques with X-ray imaging for diagnostic purposes in bile and pancreatic ducts distinct from ultrasound-guided sample collection [76].”
Please refer to the revised manuscript (pages 8-9, lines 341-348).

Reviewer #1 comments 9:
9. Full form of EUS-FNA
Response to Reviewer #1’s comment 9:
We have included “endoscopic ultrasound fine-needle aspiration” as the full form of EUS-FNA.
 Please refer to the revised manuscript (page 8, lines 3335-336).

Reviewer #1 comments 10:
10. Why the author added, “As a result, CT or MRI scans are often used for primary evaluation instead.” In the Ultrasound area.
Response to Reviewer #1’s comment 10:
We apologize to reviewer#1 for the confusion caused by the previous statement. In order to improve the flow and connection, we have added additional content to enhance the smoother transition to the upcoming sections on CT and MRI scans. 
This is the additional statement: “Additionally, alternative non-invasive imaging methods, including CT and MRI scans, offer superior image resolution for the assessment of pancreatic lesions. These imaging techniques provide enhanced detail and precision in evaluating the pancreatic tumors [76]. Consequently, CT and MRI scans are often considered for PC diagnosis as well.
Please refer to the revised manuscript (page 9, lines 344-348).

Reviewer #1 comments 11:
11. Write some lines for only CT scan.
Response to Reviewer #1’s comment 11:
We have added a supplementary content of CT scan in the introduction section 3.2.2 CT. 
This is the supplementary statement: “A conventional CT scan generates sequential three-dimensional images using a rotational and continuous X-ray technique. This method assists in detecting lesions by enhancing the contrast (CT density) between normal tissue and abnormal tumors. Furthermore, multidetector CT (MDCT) presents higher resolution images and quicker imaging duration compared to the conventional CT scans [82,83].”
Please refer to the revised manuscript (page 9, lines 351-355).

Reviewer #1 comments 12:
12. Why author add the last line about tissue biopsy in the MRI area, they already explained tissue biopsy in the next section.
Response to Reviewer #1’s comment 12:
We apologized to reviewer#1 for the confusion caused by the tissue biopsy statement in the MRI area. In order to improve the flow and connection, we have introduced additional content to enhance the smoother transition to the upcoming sections on tissue biopsy, moving it from the end of MRI section to the beginning of Tissue biopsy section.
This is the additional statement: “In addition to imaging, conducting a biopsy investigation can corroborate staging outcomes and enhance imaging-based diagnoses by integrating biological PC biomarkers. 

3.3 Tissue biopsy 
Tissue biopsy is another invasive diagnostic procedure used in clinical settings. It involves obtaining a small tissue sample from the patient’s body, which is then examined in the laboratory, typically through histological analysis, to confirm the condition of the pancreas.” 
Please refer to the revised manuscript (page 8, lines 385-393).

Reviewer #1 comments 13:
13. Explain about tissue biopsy process, not immunohistochemistry.
Response to Reviewer #1’s comment 13:
We apologize to reviewer #1 for the confusion caused by the lack statement of tissue biopsy process. We have introduced the supplementary statement of the tissue biopsy process following the statement below:
“In the context of PC, tissue biopsy involves the procedure of acquiring a sample of pancreatic tissue from a patient, intended for subsequent analysis including histological examination and immunohistochemistry. For diagnostic purposes, various tissue biopsy methodologies are available as alternatives to the conventional open surgery approach. These include laparoscopic procedures, FNA, and FNB aimed at mitigating the potential of complications, minimizing the postoperative impact, and reducing recovery time.
Laparoscopy used for diagnostic purposes involves the insertion of a camera (laparoscope) through small incisions in the abdomen. This approach allows for the localization of the pancreatic tumor, aiding in cancer staging and guiding resection decision [91]. In contrast to invasive laparoscopy, FNA and FNB are minimally invasive methodologies that entail endoscopic procedures inserted through the oral cavity to the surrounding pancreatic region. The aforementioned EUS-FNA and endoscopic ultrasound fine-needle biopsy (EUS-FNB), utilizing a larger needle, employ an endoscope equipped with an ultrasound probe to visualize the precise location of suspected lesions and to extract biopsy samples or tissue for further examination [76].”
    Please refer to the revised manuscript (page 10, lines 394-414).

Reviewer #1 comments 14:
14. The author describes Extracellular vesicles, but the heading is exosomes.
Response to Reviewer #1’s comment 14:
We have changed “Exosomes” to “Extracellular vesicles”. 
Please refer to the revised manuscript (page 12, line 516).

Reviewer #1 comments 15:
15. 130 and 131 references are not in the proper area.

Response to Reviewer #1’s comment 15:
We have changed [130,131] to [126] in the section 3.4.2 Extracellular vesicles.
Please refer to the revised manuscript (page 12, line 521).

Reviewer #1 comments 16:
16. Why do authors elaborate more about Volatile markers?

Response to Reviewer #1’s comment 16:
Thank you for your question. Due to the numerous individual technologies based on volatile markers, especially within the two main categories: analytical-based (using sensor and machines) and animal-based, it is noteworthy that the animal-based technique has not been extensively discussed in the context of pancreatic cancer detection. This presents an opportunity for us to provide a more comprehensive explanation and increase understanding of this technology within the field."

Reviewer 2 Report

This is an interesting review of approaches to pancreatic cancer detection. The review covers almost all of current methods. However, the presentation lacks coherence with some chapters written in pay language and others - in scientific language. Although all current methods are presented - no weight is given to individual approaches in terms of achievement and clinical promise. The list of citations for scientific chapters is greatly incomplete with very few references presented. For example, in protein biomarkers chapter only two studies are sited not representing the entire effort. There is no mentioning of two molecular methods (CancerSEEK and Gallery) that are currently being developed for pan-cancer screening (actually one is referred to in Ref 158, but is not named and current clinical trial is not mentioned), the requirements for screening in general and high-risk populations are not presented, there is no mentioning of differential diagnosis, some chapters, such as EV, suddenly presents some biological mechanisms, various PDAC mutations described in sfDNA chapter are not related to the molecular test that is referenced, and many others. The review is huge and needs to be significantly concentrated for readability. Illustrations are too simplistic and non-scientific. The narrative should be revised for uniformity, only relevant data should be presented. 

The review is humongous without giving any weight to different areas. The style varies from very simple targeted to laymen chapters (which were still interesting to read) to very comprehensive list of various approaches, with each part only presenting few published studies and omitting other very important reports.

Author Response

Point-by-Point Responses to the Comments Raised by the Reviewers

All reviewers:
We would like to convey our gratitude to all reviewers for their perceptive feedback on our manuscript. Their remarks have played a pivotal role in enhancing the quality of the manuscript. We have meticulously addressed each of the raised points in a detailed manner. 
Highlighted in yellow are the sections of the manuscript that have been revised. In this letter, the references to page and line numbers pertain to the updated manuscript, including Figures and Tables. We are confident that our responses effectively attend to the concerns highlighted by the reviewers, and we are optimistic that the revised manuscript will meet the standards for publication.Reviewer #2 comments to the Authorx
This is an interesting review of approaches to pancreatic cancer detection. The review covers almost all of current methods. However, the presentation lacks coherence with some chapters written in pay language and others - in scientific language. Although all current methods are presented - no weight is given to individual approaches in terms of achievement and clinical promise. The list of citations for scientific chapters is greatly incomplete with very few references presented. For example, in protein biomarkers chapter only two studies are sited not representing the entire effort. There is no mentioning of two molecular methods (CancerSEEK and Gallery) that are currently being developed for pan-cancer screening (actually one is referred to in Ref 158, but is not named and current clinical trial is not mentioned), the requirements for screening in general and high-risk populations are not presented, there is no mentioning of differential diagnosis, some chapters, such as EV, suddenly presents some biological mechanisms, various PDAC mutations described in sfDNA chapter are not related to the molecular test that is referenced, and many others. The review is huge and needs to be significantly concentrated for readability. Illustrations are too simplistic and non-scientific. The narrative should be revised for uniformity, only relevant data should be presented.

Response to Reviewer #2’s comment: 
We appreciate the comments raised by reviewer #2, and their overall positive impression of our manuscript. The suggestions and comments raised will help future research efforts. Please refer to our point-by-point responses to each comment raised.

Reviewer #2 comments 1:
1. The presentation lacks coherence with some chapters written in pay language and others in scientific language.
Response to Reviewer #2’s comment 1:
Thank you for this constructive feedback. We have thoroughly edited and streamlined the review for enhanced clarity and coherence.

Reviewer #2 comments 2:
2. All current methods are presented- no weight is given to individual approaches in terms of achievement and clinical promise.
Response to Reviewer #2’s comment 2:
This work intends to comprehensively cover the topic of pancreatic cancer diagnosis. The weight given to each part is a relative subject and largely depends on the readership’s interest. In this revised document, we have focused on balancing each section more evenly.

Reviewer #2 comments 3:
3. The list of citations for scientific chapters is greatly incomplete with very few references presented.
For example, in protein biomarkers chapter only two studies are sited not representing the entire effort. There is no mentioning of two molecular methods (CancerSEEK and Gallery) that are currently being developed for pan-cancer screening (actually one is referred to in Ref 158, but is not named and current clinical trial is not mentioned),
Response to Reviewer #2’s comment 3:
We apologize to reviewer #2. We have added more references accordingly. We have already mentioned Galleri® test and we have added CancerSEEK test, as described in the following statement below:
“Another commercial product, known as CancerSEEK, developed by Thrive, employs the MCED technique to identify cancer types by evaluating the concentrations of circulating proteins and mutation in cfDNA. The extensive clinical study involving 1,005 patients and 812 healthy controls, encompassed five types of cancer (ovary, liver, stomach, pancreas, and esophagus), and revealed a sensitivity range from 69% to 98% [148].” 
Please refer to the revised manuscript (page 13, lines 565-569)

Reviewer #2 comments 4:
4. The requirements for screening in general and high-risk populations are not presented, there is no mentioning of differential diagnosis.
Response to Reviewer #2’s comment 4:
    We have included the requirement of PC screening protocol in general and in high-risk populations following the statement below:
“Conducting a targeted examination of individuals for PC within the general population is arduous and financially unfeasible due to the inadequacy of highly precise diagnostic evaluations and the low incidence of PC. Therefore, the current screening for PC is limited to populations with potential benign tumors and those who fit the high-risk criteria recommended by the International Pancreatic Cancer Screening Consortium and the U.S. Prevention Services Task Force (USPSTF) [72-74]. Concerning the clinical presentation of PC, it is common for the afflicted individual to experience conspicuous symptoms like weight loss, pain, cutaneous indications (such as jaundice), and appreciable mass, which may not reveal any abnormalities during a physical examination [75]. 
In 2020, the American Gastroenterological Association (AGA) issued comprehensive guidelines for PC screening, with a specific focus on individuals at high risk. These high-risk groups encompass individuals afflicted with conditions such as Peutz-Jeghers syndrome, CDKN2A gene mutation, hereditary pancreatitis, Lynch syndrome, or those with a first-degree (or more) family history of PC. Furthermore, individuals with mutations in BReast CAncer gene 1 and 2 (BRCA1, BRCA2), Partner and localizer of BRCA2 (PALB2), and Ataxia-Telangiesctasia Mutated (ATM) genes are also classified as high-risk candidates. Age-specific recommendations advise individuals above 50 or below 10 years of age with familial onset to begin PC screening protocols. For those at the age of 40 with CDKN2A and PRSS1 gene mutation associated with high-risk conditions, a PC screening regimen is proposed. This entails annual PC screenings for individuals with non-PC results and a more frequent 3–6-month screening schedule for those with newly suspected PC. The AGA advocates for a multi-modal screening approach, which combines advanced imaging techniques such as a combination of magnetic resonance imaging (MRI) and endoscopic ultrasound (EUS), and the assessment of tumor markers [76]. Nevertheless, it is noteworthy that several screening/diagnostic methodologies are available to both the general population and high-risk individuals, extending beyond the techniques mentioned earlier.”
Please refer to the revised manuscript (pages 7-8, lines 276–303).

Reviewer #2 comments 5:
5. Some chapters, such as EV, suddenly presents some biological mechanisms, various PDAC mutations described in sfDNA chapter are not related to the molecular test that is referenced, and many others
Response to Reviewer #2’s comment 1:
    We agree with the comment raised by reviewer #2. We have removed the parts of a biological mechanism that were not related to the molecular test. 
3.4.1 Protein and glycan markers; Please refer to the revised manuscript (page 11-12, lines 460–504).
3.4.3 Circulating tumor cells and circulating DNA; Please refer to the revised manuscript (page 12-13, lines 536-578).
3.4.4 Epigenetic markers; Please refer to the revised manuscript (page 13-14, lines 580–613).

Reviewer #2 comments 6:
6. The review is huge and needs to be significantly concentrated for readability. The narrative should be revised for uniformity, only relevant data should be presented.
Response to Reviewer #2’s comment 6:
Thank you. We have streamlined and edited the whole manuscript for enhanced clarity and uniformity.

Reviewer #2 comments 7:
7. Illustrations are too simplistic and non-scientific.
Response to Reviewer #2’s comment 7:
We beg to differ. The targeted readership is broad, and these hand-drawn illustrations are likely to be fine by most. We note that other reviewers do not comment on this.

Reviewer #2 comments on the quality of English Language:
The review is humongous without giving any weight to different areas. The style varies from very simple targeted to laymen chapters (which were still interesting to read) to very comprehensive list of various approaches, with each part only presenting few published studies and omitting other very important reports.

Response to Reviewer #2’s comments on the quality of English Language:
We appreciate the comment raised by reviewer #2. We aimed to provide a comprehensive overview of pancreatic cancer detection test. We recognize the need for balance in covering various topics and will make necessary adjustments. However, we highlighted volatile organic compounds (VOCs) to showcase our unique contribution to the field. Your insights are invaluable in improving our manuscript’s quality. We are committed to enhancing our review based on your comments.

Reviewer 3 Report

The review “Pancreatic Cancer and Detection Methods” is interesting as it discusses various methods that are involved in the prognosis and diagnosis of the disease. The authors have done an extensive literature review and compiled this manuscript.

They have elaborately discussed the topics including the 1. Pancreas, 2. Pancreatic cancer, 3. Detection investigations, and 4. Market of the PC screening test. The review is informative, nicely written, and gives an overall idea of the topic. It is good for getting a general idea of the topic but some of the sub-topics are very lengthy and hence loosen the grip and interest. It is not necessary to write each and everything in detail.

Comments:

1.       The authors have mentioned only alcohol consumption and gallstones as the cause of pancreatitis. They should also mention other factors like the role of triglycerides in pancreatitis which is one of the major causes of the disease. Also, the authors should mention smoking.

2.       In the 1.1 section, lines 31-33, the authors mentioned that pancreatic enzymes from the exocrine cells are responsible for breaking fatty compounds. This sentence is misleading as the exocrine cells secrete various enzymes that include trypsin and chymotrypsin to digest proteins; amylase for the digestion of carbohydrates; and lipase to break down fats. Authors should correct this and add proper references.

3.       It would be better if authors add a section – “Structure of pancreas” before the function of the pancreas and marked the head, neck, and tail in Figure 1a.

4.       Is the sentence (lines 199-168) about histone modification related to PC? If yes, then add its relation to the PC. Else, it doesn’t make sense in the paragraph. Either the author should mention other types of epigenetic modifications along with this or remove the sentence if it is not related to PC.

5.       Obesity/High-fat diet has also been identified as a risk factor. Authors should add a paragraph.

6.       In Figure 3, authors should add the complete forms for the abbreviations used in the figure.

7.       As mentioned above, the authors need to shorten certain sub-topics like

a.       3.4.1 Protein and glycan markers

b.        3.4.3 Circulating tumor cells and circulating DNA

c.       3.4.4 Epigenetic markers

d.       3.4.5 Volatile markers

e.       All the subtopics under Volatile markers

The English language is fine. Some minor corrections needed.

Author Response

Point-by-Point Responses to the Comments Raised by the Reviewers

All reviewers:
We would like to convey our gratitude to all reviewers for their perceptive feedback on our manuscript. Their remarks have played a pivotal role in enhancing the quality of the manuscript. We have meticulously addressed each of the raised points in a detailed manner. 
Highlighted in yellow are the sections of the manuscript that have been revised. In this letter, the references to page and line numbers pertain to the updated manuscript, including Figures and Tables. We are confident that our responses effectively attend to the concerns highlighted by the reviewers, and we are optimistic that the revised manuscript will meet the standards for publication.

Reviewer #3 comments to the Author
The review “Pancreatic Cancer and Detection Methods” is interesting as it discusses various methods that are involved in the prognosis and diagnosis of the disease. The authors have done an extensive literature review and compiled this manuscript.

They have elaborately discussed the topics including the 1. Pancreas, 2. Pancreatic cancer, 3. Detection investigations, and 4. Market of the PC screening test. The review is informative, nicely written, and gives an overall idea of the topic. It is good for getting a general idea of the topic but some of the sub-topics are very lengthy and hence loosen the grip and interest. It is not necessary to write each and everything in detail.

Response to Reviewer #3’s comment: 
We are pleased to hear that you found our review interesting and informative. In response to your comments, we have revised the depth of certain sub-topics and balanced of key. Your insights are valuable in refining our manuscript.

Reviewer #3 comments 1:
1. The authors have mentioned only alcohol consumption and gallstones as the cause of pancreatitis. They should also mention other factors like the role of triglycerides in pancreatitis which is one of the major causes of the disease. Also, the authors should mention smoking.

Response to Reviewer #3’s comment 1:
We agree with the comment raised by reviewer #3. We have added the other risk factors of pancreatitis following the statement below:
“Among the most prevalent pancreas diseases is acute pancreatitis, an inflammatory condition triggered by factors such as alcohol consumption, gallstones, type 2 diabetes, smoking onset, and hypertriglyceridemia [2,3]. The incidence of acute pancreatitis varies widely, with estimates ranging from 10-40 cases per 100,000 population, with a tendency towards higher rates in regions with high alcohol consumption [3]. Gallstones, which may dislodge from the gallbladder and obstruct the bile duct, are another common cause of acute pancreatitis, particularly in the elderly population. The obstruction of the bile duct and pancreatic duct, which converge at the ampulla of Vater, leads to pancreas inflammation. Additionally, type 2 diabetes has been associated with elevated risk of gallstone [2]. 
Smoking is associated with tissue inflammation and fibrosis, when coupled with heavy alcohol consumption, can incite severe inflammation, rendering pancreatic cells more susception to autodigestion [4,5]. Hypertriglyceridemia is characterized by an elevated serum fatty acid concentration, which can inhibit mitochondrial injury in pancreatic acinar cells, culminating in inflammation, and a consequential reduction in pancreatic duct function [6].”
Please refer to the revised manuscript (page 3, lines 61-75).

Reviewer #3 comments 2:
2. In the 1.1 section, lines 31-33, the authors mentioned that pancreatic enzymes from the exocrine cells are responsible for breaking fatty compounds. This sentence is misleading as the exocrine cells secrete various enzymes that include trypsin and chymotrypsin to digest proteins; amylase for the digestion of carbohydrates; and lipase to break down fats. Authors should correct this and add proper references.

Response to Reviewer #1’s comment 2:
We have revised this statement following the statement below:
“The majority of pancreatic cells are exocrine cells, which make up the exocrine glands and ducts. These cells are responsible for secreting a range of enzymes including amylase for carbohydrate digestion, proteases such as trypsin, chymotrypsin, elastase, and carboxypeptidases for amino acids digestion, and lipase for the breakdown of fatty acids.”
Please refer to the revised manuscript (page 1, lines 38-42).

Reviewer #3 comments 3:
3. It would be better if authors add a section – “Structure of pancreas” before the function of the pancreas and marked the head, neck, and tail in Figure 1a.

Response to Reviewer #3’s comment 3:
We have added the structure of Pancreas following the statement below:
“1.1 Structure of the pancreas
A pancreas is a visceral organ located posterior to a stomach in the form of a leaf. It measures roughly 15 centimeters in length and 5 centimeters in width in adults. The pancreas is subdivided into 4 distinct regions: a head, neck, body, and tail. The head of the pancreas is the broader part, situated adjacent to the stomach and duodenum junction, where the bile duct either lies within a groove on its surface or passes through its substance. The neck and body regions are surrounded by major blood vessels. Finally, the tail of the pancreas is located anterior to the left kidney and spleen [1].”
Please refer to the revised manuscript (page 1, lines 27-34).

Reviewer #3 comments 4:
4. Is the sentence (lines 199-168) about histone modification related to PC? If yes, then add its relation to the PC. Else, it doesn’t make sense in the paragraph. Either the author should mention other types of epigenetic modifications along with this or remove the sentence if it is not related to PC.

Response to Reviewer #3’s comment 4:
 We have rewritten these sentences to make them more easily understandable following the statement below:
“Epigenetic changes refer to modifications to DNA that impact gene expression without changing the underlying DNA sequence. These changes are often associated with specific genes related to histone modification enzymes. The histone modification enzymes, including mixed-lineage leukemia (MLL2/3) histone methylases, lysine demethylase 6A (KDM6A), and histone methyltransferase, are related to histone modification and chromatin-regulating genes. These alterations promote uncontrolled cell growth and contribute to PC development [34,39,40].”
Please refer to the revised manuscript (page 5, lines 186–192).

Reviewer #3 comments 5:
5. Obesity/High-fat diet has also been identified as a risk factor. Authors should add a paragraph.

Response to Reviewer #3’s comment 5:
We have included this statement following the statement below:
“…and high-calorie diet…
…Thus, obesity and high-fat diet are proposed inducers of tissue inflammation in the development of PC.”
 Please refer to the revised manuscript (page 6, lines 215, and 218-219).

Reviewer #3 comments 6:
6. In Figure 3, authors should add the complete forms for the abbreviations used in the figure.

Response to Reviewer #3’s comment 6:
We have added the full forms of the abbreviations in Figure 3. 
Please refer to the revised manuscript (page 14, revised Figure 3).

Reviewer #3 comments 7:
7. As mentioned above, the authors need to shorten certain sub-topics like

a.       3.4.1 Protein and glycan markers
b.       3.4.3 Circulating tumor cells and circulating DNA
c.       3.4.4 Epigenetic markers
d.       3.4.5 Volatile markers
e.       All the subtopics under Volatile markers

Response to Reviewer #3’s comment 7:
We have condensed each section, particularly the one on volatile markers. Nevertheless, given the multitude of technologies centered around volatile markers, with a specific focus on the two primary categories—analytical methods (utilizing machines) and animal-based approaches—it's essential to highlight that the animal-based technique remains relatively underexplored in the realm of pancreatic cancer detection. This provides us with an opportunity to deliver a more thorough exposition, enhancing comprehension of this technology within the field. Please check the following statement below:
3.4.1 Protein and glycan markers; Please refer to the revised manuscript (pages 11-12, lines 470–514).
3.4.3 Circulating tumor cells and circulating DNA; Please refer to the revised manuscript (pages 12-13, lines 546-593).
3.4.4 Epigenetic markers; Please refer to the revised manuscript (pages 13-14, lines 595–628).
3.4.5 Volatile markers; Please refer to the revised manuscript (pages 15-19, lines 635–860).

Reviewer #3 comments on the quality of English Language:
The English language is fine. Some minor corrections needed.

Response to Reviewer #3’s comments on the quality of English Language:
We have corrected some minor mistakes. We appreciate your feedback on the quality of English language.

Round 2

Reviewer 3 Report

The revised manuscript “Pancreatic Cancer and Detection Methods” has been significantly changed in accordance with the suggestions made to the authors.

The modified manuscript is suitable for publication in the journal.

The English language is fine. Authors are requested to carefully go through the manuscript to remove some minor errors.